# Evaluation of a Program of Aquatic Motor Games in the Improvement of Motor Competence in Children from 4 to 5 Years Old

**DOI:** 10.3390/children9081141

**Published:** 2022-07-29

**Authors:** Juan Ángel Simón-Piqueras, Alejandro Prieto-Ayuso, Elena Gómez-Moreno, María Martínez-López, Pedro Gil-Madrona

**Affiliations:** Departamento de Didáctica de la Educación Física, Música y Plástica, Facultad de Educación, Universidad de Castilla-La Mancha, 02071 Albacete, Spain; alejandro.prieto@uclm.es (A.P.-A.); elena.gomez18@alu.uclm.es (E.G.-M.); maria.martinez71@alu.uclm.es (M.M.-L.); pedro.gil@uclm.es (P.G.-M.)

**Keywords:** preschool education, aquatic environment, skills, methodology, curriculum, swimming

## Abstract

The present study aimed to verify whether a program of initiation to the aquatic environment composed of motor games (motor stories, learning environments and motor circuits) is more effective in the acquisition of aquatic skills in early childhood education than a traditional program. For this purpose, the Escala de Competencia Motriz Acuática for 4–5 year-old children was used before and after both programs. This scale has two factors, one named Familiarization, linked to more elementary aquatic abilities, and the Immersion factor, which integrates more advanced aquatic abilities. The two programs were used as independent variables, and there was a sample of 17 children over six weeks. The results show that both programs improved the acquisition of aquatic skills. The program that integrated motor games impacted the improvement of skills related to the Immersion factor significantly more.

## 1. Introduction

Physical Education (PE) in early childhood education has integrated the concepts of psychomotor skills with the contributions of different sciences (psychology, pedagogy, sociology, etc.) in order to develop more effective techniques and intervention methods in the educational system [1]. In this way, mere physical training has been omitted, and the focus has shifted towards different approaches centered on the student’s interaction with his/her environment as a method of reciprocal stimulation [2], where learning tasks are adjusted to the levels of psychomotor development of the child [3], and activities are planned in which the student manages their own learning [4]. This evolution of approaches has resulted in the development of new methodologies adapted to these tendencies where motor play takes on important relevance. Some of these methodologies are motor circuits, motor stories, learning corners and learning environments [5]. 

Currently, the role of the aquatic environment is highlighted as a relevant educational tool that allows the student to relate the activities performed in the pool with those performed in the classroom under a comprehensive school approach [6]. Thus, the aquatic environment is presented as an optimal scenario to work on the global development of the individual [7,8], suitable due to its characteristic of simulating, through motor action, the knowledge of one’s own body [9] and its relationship with the aquatic environment [10]. However, despite this enormous potential, this educational content is not yet implemented in the educational system [11].

Delving deeper into the specific literature on the use of aquatic development in the ages included in early childhood education, it can be appreciated that the concept of physical training is still maintained [12], or in the best of the cases, a trend linked to the exploration of the environment and the positive effect for the student [13]. Thus, the application of the different approaches and methodologies mentioned at the beginning of this article in the aquatic environment with an educational purpose has rarely been applied and studied. 

In the few existing studies on the educational role of the aquatic environment, a distinction is usually made between traditional and recreational approaches [10], including those approaches that include current educational methodologies in the recreational field such as those proposed here, based on motor play and relegating the role of activities based on motor repetition to traditional approaches. Within the playful proposals, those that propose the achievement of objectives, implemented under playful and relaxed environments, and recommend the use of attractive teaching materials for students can be included [13].

The strength of the methodologies that promote motor games includes and goes beyond the playful aspect of these methodologies. In relation to motor stories, this methodology consists of a story that takes the students to an imaginary scenario in which the characters cooperate with each other, within a context of challenge and adventure, to achieve a shared objective with which the children can identify. The story gives rise to proposals in which the students participate, from motor activity, emulating the characters of the story itself [7]. Conde-Caveda and Viciana [14] consider that these stories enhance the behaviors of imitation, research, experimentation, etc. Moreno, Huescar, Polo, Carbonell and Messeguer [15] highlight the enormous potential of symbolism through motor stories and its role as an educational tool.

Another methodology used in the motor game program is the learning environments, which are scenarios for the creation and construction of new learning, in which the educator intentionally creates a series of activities to achieve learning objectives [5]. The learning environment itself is defined in different ways: as a meeting place and a dynamic and inter-relational space that develops from mobile and dynamic relationships which are transformed and modified, inviting the complexity of the actions of the people who inhabit it, both adults and children [16]. It is composed of "physical, social, cultural, psychological, pedagogical, pedagogical, human, historical elements, among others, that are presented related to each other and that favor, or hinder, interaction, relationships, identity, and sense of belonging" [17]. In relation to Physical Education [18], learning environments are defined as the organization of spaces and materials in such a way that new motor patterns arise spontaneously or previous patterns are strengthened. They are spaces in which students are in continuous movement, both to access different proposals and to carry them out [19,20]. Finally, the motor circuits consist of the distribution of objects in the space in a way that stimulates motor action in a varied way, allowing the student to be the one who ventilates this motor action through the game [1,2,3,4,5,6,7].

Delving into the documentation referred to the use of current educational methodologies in the aquatic environment, it is possible to verify that, in relation to motor stories, there are some work proposals based on these [7,21,22,23]; however, research on the results of their use is scarce. For example, Moreno et al. [15] analyzed whether a program based on symbolic activities (motor stories) was shown to be more effective than a traditional program in terms of the development of students’ perceived ability and actual ability. The results showed that although the students who carried out the program based on motor stories were perceived to be more capable, there were no significant differences in actual ability. On the other hand, Navarro, Sánchez and Simón [7] used a program of aquatic motor stories to analyze the teachers’ opinion on the implementation of this methodology in educational swimming classes, obtaining the methodological aspects and its attitudinal effects on the students as the most relevant positive elements, and the greatest difficulty in its implementation and organization as negative aspects. 

If research in the aquatic environment on motor stories is scarce, it can be considered almost null when we refer to other methodologies such as learning environments or learning corners. There are some contributions of work in the aquatic environment [13,14,15,16,17,18,19,20,21,22,23,24], but beyond these contributions, it is difficult to locate research that allows us to make concrete conclusions in the aquatic environment. In this sense, research is focused on evaluating punctual methodologies, such as studies on motor stories, when the reality of current PE poses the integration of such methodologies simultaneously with the use of other traditional methodologies. For this reason, the evaluation of a program of initiation to the aquatic environment that interrelates these methodologies to assess their results on aquatic motor competence seems evident.

Thus, the present study aims to determine whether methodologies based on motor games are more efficient in the development of aquatic motor competence in children aged 4 and 5 years than methodologies based on motor repetitions.

## 2. Materials and Methods

### 2.1. Design

This was a quasi-experimental design because, although pre and post measurements were taken in two groups, the sample was incidental [24]. Two community-based aquatic initiation programs were used as independent variables, one with aquatic methodologies based on the repetition of motor tasks, and a second program that included adaptations of methodologies based on motor games. The objective was to measure improvements in aquatic motor competence. For this, the two factors of the Aquatic Motor Competence Scale (ECMA), designed and validated by Moreno [25], were used as dependent variables.

### 2.2. Participants

Seventeen students between 4 and 5 years old (M = 4.95, SD = 0.75) were selected incidentally for ease of access to participate. Of the participants, 41% percent were boys and 59% girls. The motor games group (Mgg) consisted of 8 students, and the repetition motor group (Rmg) consisted of 9 students. None of the students had previous experience in an aquatic initiation program. The size of the sample was analyzed using the Hopkins Sample-size Estimation for Magnitude-Based Decisions method [26,27,28] using studies with samples, applications and methodologies very similar to those of this research, published in impact journals [28]. The results obtained allowed us to conclude that the sample, despite being small, may have been adequate. Despite this, Hopkins points out that in sample sizes of less than 10 subjects per group, errors can occur. However, it should be noted that there are studies on this subject that are similar in terms of sample size and group distribution [28].

The study was carried out in the municipal swimming pool of a town in the province of Albacete, where all the participants in the study lived. The participants attended swimming lessons during after-school hours. All the parents of the participants signed the corresponding informed consent forms, and the study was approved by the ethics committee of the sports board and the corresponding federation (FNCLM).

### 2.3. Instrument

The instrument used was the Aquatic Motor Competence Scale (Escala de Competenica Motriz Acuática, E.C.M.A), designed and validated by Moreno [25], in its version focused on children aged 4–5 years. The scale is composed of 16 items, and the response to it is dichotomous (Yes or No), valued in subsequent statistical analyses as 1 (No) and 2 (Yes). This scale has a total internal consistency of 0.92 (Cronbach alpha) [25].

This scale evaluates two factors called Familiarization and Immersion. The first contains nine items that assess aspects related to the attitude shown by the children in contact with the aquatic environment, for example: *Does not cry in front of the water, Moves holding on to the edge, Is able to pick up objects of different sizes and bring them to the edge, etc...* which are also the fundamental aspects of the initial classes in order to explore and become aware of the new environment, which is considered the first stage in the initiation to it [12]. This factor has total internal consistency of 0.92 (Cronbach alpha). 

As for the Immersion factor, it contains seven items referring to more advanced aquatic motor skills related to respiratory control, flotations in different positions and displacements related to actions with the head in immersion; some examples of them are: *performs an underwater exhalation, picks up objects from the deep pool without help or with impulse on the wall slides in supine lying*. This factor has total internal consistency of 0.85 (Cronbach alpha). 

This instrument was completed by a single researcher on two occasions, the first at the beginning of the program (pre), and the second at the end of the program (post). 

### 2.4. Protocols

Firstly, the implementation of the study was made known to the local town council in order to obtain the corresponding ethical permissions from those responsible for the municipal swimming pool. Informed consent was given to the parents at an informative meeting in which the objectives of the program and the necessary requirements of the researcher were explained to them. Secondly, participants were selected incidentally and randomly classified into an (Rmg) group and an (Rmm) group. Thirdly, the programs were implemented for the (Rmg) group, based on traditional methodologies centered on motor repetitions, and the (Rmm) group using methodologies based on motor stories. Prior to the implementation of the programs, an initial evaluation (pre) was performed, which was followed by the implementation of the programs and an evaluation at the end of the program (post). 

The duration of both programs was six weeks, with two one-hour sessions per week. In the (Rmm) group, all the sessions were planned with aquatic motor activities in the traditional way. In the experimental group, a traditional session with repetitive aquatic motor activities was alternated with other sessions in which other types of educational methodologies were used (2 motor stories, 1 motor circuit and 1 learning environment) based on motor games. The total duration of the study was six weeks, including the initial and final evaluations. An example of the development of the different activities is presented in Table 1 under each of the proposed methodologies.

For the calculation of the sample size, the Hopkins estimation application was used. The SPSS v. 20.0 statistical program was used. The Shapiro–Wilk test was performed to check if the sample followed a normal distribution pattern, showing that the sample did not follow such a pattern (*p* < 0.05); therefore, the use of non-parametric tests was required for the subsequent analyses. For this reason, the Wilcoxon signed-rank test was used, with the purpose of comparing the pre and post measurements of each group, and the Mann–Whitney U test was used with the objective of finding out if there were significant differences between the control and experimental groups after the application of both programs. Finally, the effect size and error were calculated using the G*Power 3.1 application.

## 3. Results

First, the preliminary analyses using the non-parametric test for independent samples (Mann–Whitney U) did not reveal significant differences in the Familiarization factor (U = 20.00, *p* > 0.05) or in the Immersion factor (U = 25.00, *p* > 0.05), so we started with homogeneous groups.

At the end of the programs, the statistical analysis was performed to check the differences between the groups themselves, applying the Wilcoxon test to determine the differences in the factors Familiarization with the aquatic environment (Table 2) and the Immersion factor (Table 3).

Within the motor games group, both the Familiarization factor with the aquatic environment and the Immersion factor showed significant differences in the results of the pre and post evaluation. In relation to the repetition motor group, the Familiarization factor with the environment showed statistically significant differences between the initial and final evaluation, but this was not the case for the Immersion factor, as no significant differences were obtained. Secondly, to evaluate which program was more effective, the differences between the control group and the experimental group were analyzed. For this purpose, the Mann–Whitney U test was used. The results, as can be seen in Table 4 and Table 5, show that no statistically significant differences were obtained in the aquatic environment Familiarization factor. However, in the Immersion factor, significant differences were obtained between the motor games group and the repetition motor group, the first being the one that obtained better results.

## 4. Discussion

The purpose of this study was to verify the existing differences in aquatic motor competence in children in early childhood education through the implementation of a program of initiation to the aquatic environment that included current educational methodologies with motor games (motor stories, learning environments and motor circuits), as opposed to a traditional program with repetition motor games. In this way, through an aquatic motor skills evaluation instrument, the aim was to find out which program was more effective in achieving mastery of the skills evaluated. 

Several authors consider the mastery of the aquatic environment at an early age a procedure linked to the overall development of the individual [29,30,31,32], so it can be considered that the development of aquatic motor skills is the result of the acquired knowledge of the characteristics of the environment where students develop, of knowing how they should use their body in an effective way in that environment, increasing their autonomy [7], so a program of initiation to the aquatic environment that incorporates motor games methodologies should be shown, which is more efficient as a tool for the acquisition of such motor skills.

The results obtained show on the one hand that both programs have been effective as a tool for aquatic skills linked to the Familiarization and immersion factors of the instrument used, showing significant differences between the initial and final evaluations. However, when the results of both methodologies were compared, disparity results were obtained, since on the one hand no significant differences were obtained between both groups in relation to the Familiarization factor, but on the other hand, it was found that the motor game group was more effective than the repetition motor group.

The reason for the disparity in these results may be because the skills included in the aquatic environment Familiarization factor are simple skills, and the duration of the program was sufficient to acquire them with both methodologies. However, due to the complexity of the skills included in the Immersion factor, they were acquired more effectively in the group that used motor games. In this sense, the literature states that the skills included in the Familiarization factor are what must first be worked on and mastered in the aquatic environment [12,33,34,35]. Therefore, it is possible that, despite the absence of significant differences between the groups, the experimental group acquired the skills included in the Familiarization factor more quickly, and this allowed them to progress more effectively in the skills included in the Immersion factor, which are skills of greater complexity, and which include the mastery of the skills included in the Familiarization factor. 

Proof of the above-described concept is the important role that is generally given to the adequate mastery of skills related to awareness of the aquatic environment for the acquisition of more complex skills [12,36,37]. In this sense, this importance is not only affected from the motor point of view, but from other aspects such as confidence, autonomy and competence [38]. A student who is perceived as competent will be more autonomous, and the consequence is that he/she explores and interacts with the environment in a safer and more confident way [7,8,15,39].

The use of the motor game methodology is likely to have encouraged the exploration of the environment and by the logical relationships that are favored through interactions with objects, with the environment, with other people and with oneself [15], allowing students to focus their attention on the experiential processes of playful activities [8,39], encouraging the students’ construction of their own learning, provoking decision making, observation, experimentation [13], inducing students to maintain a more receptive attitude towards the program and increasing their motivation and enjoyment during the activity [7].

The main contribution of this work is that it is one of the first studies that analyzes the effect of integrating different current methodologies simultaneously in a program of initiation to the aquatic environment. However, despite the benefits of the study, some limitations also arose, such as the small sample of the study, previously commented on, or the duration of the program.

Finally, as prospective research, it would be necessary to continue to carry out studies that focus on the motivation of children in early childhood education according to the educational program used (traditional or current). On the other hand, it is necessary to compare these same educational methodologies to check which of them improve aquatic motor skills. Another line of research that emerges from this study is to check whether the use of current educational programs in the initiation to the aquatic environment, improve not only the skills of swimming but also other aspects of development such as cognitive or social aspects.

## 5. Conclusions

Finally, in line with the research objectives of this work, it is concluded that the aquatic initiation program that integrates educational methodologies based on motor games is more effective than a traditional aquatic initiation program based on motor repetitions.

## Figures and Tables

**Table 1 children-09-01141-t001:** Example of the development of the different activities.

Motor Games Group				Motor Repetition Group
Aquatic Skill	Motor Stories	Learning Environments.	Motor Circuit (1 Station of the Circuit)	Motor Repetition
Diving and respiratory apnea	Fragment of the motor story: "The children arrived at the place where the fish Dori was with other fish, these were stuck in a cage and could not get out, only with the help of the children could open the cage, but for this they had to collect some keys from the bottom to open it to free Dori and the rest of the fish (The children will get out of the pool and jump to catch different rings from the bottom of the pool to be thrown)”.	The teacher will introduce into the pool a variety of different materials that float on the surface, sink and remain semi-submerged. The number of materials, colors and shapes should be motivating. Students will be allowed to explore the environment, and students are encouraged to use free play in the environment.	Mirror Game: The children dive underwater and one of them makes a face underwater. The others have to guess what grimace he/she made.	Colored hoops game: A group of students must work together to collect hoops from the bottom of the pool and sort them by color at the edge.
Diving to the bottom of the pool			Colored hoops game: A group of students must work together to collect hoops from the bottom of the pool and sort them by color at the edge.	Dive to the bottom of the pool, touch it with your hand and come to the surface.

**Table 2 children-09-01141-t002:** Wilcoxon test applied to Factor 1 (Familiarization).

Wilcoxon Test					
		Average	DT	Z	*p*-Value
Factor 1	Mgg pre	1.5	0.13	−2.588	0.01 *
	Mgg post	1.87	0.39		
	Rmg pre	1.58	0.93	−0.2716	0.007 *
	Rmg post	1.80	0.11		

* (*p* < 0.01) Mgg: motor games group. RMG: repetition motor group.

**Table 3 children-09-01141-t003:** Wilcoxon test applied to Factor 2 (Immersion).

Wilcoxon Test					
		Average	DT	Z	*p*-Value
Factor 2	Mgg pre–post	1.05	0.92	−2.460	0.014 *
	Mgg post	1.52	0.26		
	Rmg pre	1.11	0.11	−0.1000	0.317
	Rmg post	1.16	0.13		

* (*p* < 0.01) Mgg: motor games group. RMG: repetition motor group.

**Table 4 children-09-01141-t004:** Mann–Whitney U test results.

	Familiarization (F1)	Immersion12345(F2)
Mann–Whitney U	23	8.5
*p*-Value	0.119	0.007 *

* (*p* < 0.01) motor games group. GC: repetition motor group.

**Table 5 children-09-01141-t005:** Effect size and error probabilities.

		Effect Size	*p*	Prob Error
Factor 1	Mgg pre–post	0.93	0.01	0.95
	Mrg pre–post	0.92	0.01	0.81
Factor 2	Mgg pre–post	1.74	0.01	0.93
	Mgg-Mrr	1.75	0.01	1

## Data Availability

The data presented in this study are available on request from the corresponding author. The data are not publicly available due to privacy.

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
