# Peer review of "Evaluation of a Program of Aquatic Motor Games in the Improvement of Motor Competence in Children from 4 to 5 Years Old"

_children, 2022, doi:10.3390/children9081141_

Round 1

Reviewer 1 Report

Dear Editor,

Thank you for the opportunity to review this article. In general, the aim of the study is clear: comparing two methods, motor games and motor repetition, towards the acquisition of aquatic skills in children. However, I think the protocol administered specifically is lacking, as I do not understand how it would advance the literature if at least an example protocol of the proposed activities is not clearly provided. I would suggest adding a table in the Method section, “Protocols” paragraph, differentiating the two methods better. Finally there are some clarifications listed below:

Line 60: "includes" instead of "include"

Line 63: "and" instead of "y" 

Line 76: after Sanchez remove comma ","

Line 95: there is an extra comma "," to remove

Line 99: "was to measure" instead of "was measure"

Line 94: Materials and Methods. The method is very confusing. You should break it down into subsections and then according to your information:

1)     Design

2)     Participants

3)     Tool /Instrument

4)     Protocols. As mentioned before, add a table with the two protocols.

5)     Statistical analysis

Line 104: why "Forty-one percent were boys" is written in extended and "59% girls" in percentage? Use one form for everything. I suggest writing percentages in numerical form for better understanding and readability.

Line 115: if possible isert a citation.

Line 112: it is not clear what ECMA means if you write the name of the scale in English. You should also include the original name Las Escalas de Competencia Motriz Acuática (ECMA).

Line 119: when you describe the questionnaire items you could put the colon and write Item 1) Does not cry in front of the wate, Item 2) and so on.

Line 112: insert a citation if possible

Line 124: "The immersions factor contains" instead of "As for the immersions factor, it contains"

Line 126: "with the head in immersion. Some examples are: Item 1) performs an underwater exhalation, Item 2) and so on."

Line 128: this seems like a repetitive sentence since you already stated in line 124 that these skills are more advanced.

Line 129: insert a citation if possible.

Line 131: "This instrument was completed by a single researcher on two occasions:"

Line 133: "Firstly" instead of "First"

Line 134: "in order to obtain the" instead of "with the purpose of obtaining"

Line 138: "Thirdly" instead of "Third"

Line 138 - 140: previously you nicknamed the experimental group (mmg) and control group (rmg) so now you should use the abbreviations. The same is true in the next paragraph 144 to 150.

Line 155: comma "," instead of semicolon ";"

Line 165: "perfomed" instead of "carried out"

Line 172: "(immersion)" instead of "(inmersions)"

Line 175: use the abbreviation of motor games group (Mgg). Adjust throughout the manuscript also for Rmg.

Line 166 - 167 and Line 180 - 182: this sentence is already repeated in the method. Synthesize for example "from the comparison of the two groups...."

Line 199 - 205: I think this paragraph should be inserted later, giving priority to the summary of the results obtained.

Line 207: delete "on the one hand"

Line 210: "However, when the results of both methodologies were compared, disparity results were obtained. On one hand, no significant differences were obtained between both groups...."

Line 226 - 227: reword the sentence because it is not clear.

Line 241: add "first studies to our knowledge", unless you are 100% sure this is one of the first study in literature.

Line 268: the style of the references needs to be changed. Consult the authors' guidelines.

Example

Journal Articles:
1. Author 1, A.B.; Author 2, C.D. Title of the article. Abbreviated Journal Name YearVolume, page range.

1.     Moreno-Murcia, J.A. Desarrollo y validación preliminar de escalas para la evaluación de la competencia motriz acuática en escolares de 4 a 11 años. RICYDE. Revista Internacional de Ciencias Del Deporte 2005, 1(1), 14-27. http://dx.doi.org/10.5232/ricyde2005.00102

Good work!

Author Response

Please see the attachment below with the major changes.

Reviewer 2 Report

Interesting work. 

Please find my comments in the attached file. 

Major comments:

- related to the inferences, it is approriate to clarify the sample size calculation in advance of the study. If not performed, it should be discussed thouroughly in the discusssion section. you may use this link as a guide towards sample size calculations and potential errors that have been made: https://www.sportsci.org/2006/wghss.htm 

- the motor games program should be described in more detail, so that your research is more transparent 

- please provide some more comprehensive information on the Etical approval in the methodology in one paragraph. The information has been spread over several sections and that is confusing or attracts attention.

Minor comments:

please correct typo's and English writing errors

Author Response

First and foremost, we want to thank you for your constructive review. Thank you very much.

We want to indicate that:

  1. We have corrected the typo's and English writing errors.
  2. We have added information about the different methodologies, especially those of the group of engine games. We have also added a table exemplifying them
  3. We have added information about the ethics committee.
  4. Regarding the size of the sample and its calculation, we have used as a guide and referenced the information you sent us. Carrying out the calculations, the results are positive, but it should be noted that Hopkins himself indicates that below 10 subjects per group there may be errors. In this sense, we must comment that we cite similar studies with similar groups published in indexed journals and some included in JCR

Thank you for the information you sent us and it is a pity that the period to resubmit the article is so short and does not allow us to go deeper, but it will be very useful in the future.

Thanks for all.

Round 2

Reviewer 1 Report

All changes have been made. Good work!

Reviewer 2 Report

The comments and suggestions have been addressed properly. The clarifying table on the content of the approach that has been used is of great additional value for the readership. Congratualations. 

Some minor comment:

Line 72: Typo - (L)learning environment.

This manuscript is a resubmission of an earlier submission. The following is a list of the peer review reports and author responses from that submission.

Round 1

Reviewer 1 Report

a quasi experimental design for this investigation is appropriate. However, the limitations of the numbers, although reported, are not fully taken into account when reporting on the implications of the research and the conclusion should be made much more tentative in light of this limitation. The study as it stands seems to be appropriate for a pilot, but I question its effectiveness as a full blown study.

Sometimes the technical language needs to be more fully explained- for example, it would be useful to explain briefly what terms such as motor-stories, learning corners, learning environment mean in the context of the study, as well as  what is considered as traditional, beyond describing it as repetition. 

There are some points that need clarification. For example p2 lines 54-58. Do you mean that most authors recommend a playful approach to the acquisition of aquatic skills, or the opposite?  

It would be helpful to understand the context of the review of literature- references seem quite sparse, which may be due to the lack of relevant studies, but it would also be helpful perhaps to situate the study in the wider context of PE, which would be helpful in relationship to the fit for this journal.

Reviewer 2 Report

My understanding is that this study compares outcomes of two community-based aquatic programs, both of 4 weeks duration.  The two programs are referred to as experimental and control, but there is no rationale for applying these labels and the two programs should be referred to as interventions or community-based programs.  There is a pre- and posttest of aquatic motor competence.  No information on reliability, validity or any similar metrics is provided. It doesn’t seem there are any other data collected that might indicate comparability (or lack of comparability) of the participants in each group.  It doesn’t appear that the pretest scores of participants from the two programs have been compared to determine comparability.

I wasn’t able to find any information regarding the Escala de Competencia Motriz Acuátic so I am not sure how it has been scored.  If there are 16 items and 2 factors, my guess was that there are 8 items for each factor with yes being scored 1 and no being scored 0.  If that’s correct, the post-test averages suggest that neither program is particularly helpful as the change is very small, even if it is statistically significant. Another possibility is that there is a serious floor effect for the Escala de Competencia Motriz Acuátic. There isn’t enough information provided to determine if the significant differences are due to changes in one or two participants rather than a true group change.

This is an interesting pilot study, but it is too small and has too many difficulties for publication.  The groups cannot be compared as there isn’t any information to justify a direct comparison.  There are not enough participants.  Also, with just one group in each program it is impossible to take into account any cluster effects.